# Production of Methanol on PdCu/ATO in a Polymeric Electrolyte Reactor of the Fuel Cell Type from Methane

Camila M. Godoi, Isabely M. Gutierrez, Paulo Victor R. Gomes, Jessica F. Coelho, Priscilla J. Zambiazi [ID], Larissa Otubo, Almir O. Neto and Rodrigo F. B. de Souza *[ID]

Instituto de Pesquisas Energéticas e Nucleares, IPEN-CNEN/SP, Cidade Universitária, Av. Prof. Lineu Prestes, 2242, São Paulo 05508-900, SP, Brazil
* Correspondence: souza.rfb@gmail.com

**Abstract:** The search for alternatives for converting methane into value-added products has been of great interest to scientific, technological, and industrial society. An alternative to this could be the use of copper-doped palladium catalysts with different proportions supported on metal oxides, such as $Sb_2O_5.SnO_2$ (ATO) catalysts. These combinations were employed to convert the methane-to-methanol in mild condition using a fuel cell polymer electrolyte reactor. The catalysts prepared presents Pd, CuO, and $Sb_2O_5.SnO_2$ phases with a mean particle size of about 9 nm. In activity experiments, the $Pd_{80}Cu_{20}$/ATO indicated maximum power density and maximum rate reaction for methanol production when compared to other PdCu/ATO materials. The use of ATO as a support favored the production of methanol from methane, while PdCu with high copper content demonstrated the production of more oxidized compounds, such as carbonate and formate.

**Keywords:** methane to products; polymer electrolyte reactor; fuel cell reactor





## 1. Introduction

Within the growing concern about greenhouse gas emissions, methane has an ambiguous role; if released into the atmosphere it is a more dangerous greenhouse gas than $CO_2$, but it can be used as an alternative to petroleum because it burns more efficiently than petroleum derivatives, although it emits $CO_2$ [1–4].

However, transforming the most stable of hydrocarbons due to their difficulty in polarizability and high binding energy, into other chemical species, such as methanol, formaldehyde, formic acid, and ethane, among others, all of them having industrial applications, can be an interesting method [5–7]. The primary route currently adopted is by thermal processes such as Fischer–Tropsch, which has problems with high power consumption, energy dissipation to the environment, and low yield [8–12]. In recent times, other technologies for conversion have been studied, such as photochemical [13–16] and electrochemical pathways, which can lead to partial oxidation of this gas in mild conditions.

The electrochemical route can be a good technological option for being able to select the desired product by the amount of energy to which the system is submitted [17], which can be performed in a faradaic manner and depends on the direct injection of energy to break the C-H bonds of methane [18]. Conversely, the energy required is higher than the process occurring indirectly, which depends on the activation of water to obtain reactive oxygenated species such as the •OH, which reacts with the hydrocarbon [19], as in Equations (1)–(4):

$$CH_{4(gas)} + M_{(ads)} \rightarrow M\text{-}CH_{4(ads)} \tag{1}$$

$$H_2O_{(liq)} + M \rightarrow MOH + H^+ \tag{2}$$

$$MOH \rightarrow M + OH\text{-} \text{ or } \bullet OH \tag{3}$$

$$\bullet OH + CH_{4(gas)} \rightarrow CH_3OH \tag{4}$$

The electrochemical method still allows the co-generation of electrical energy along with the conversion of methane into methanol; if the process is performed in fuel cell reactors [19], these systems present the advantage of operating at mild conditions and in continuous flux. De Souza et al. [20] demonstrated that the catalyst that obtained the highest conversion of methane to methanol was based on Pd and Cu with rate reaction for methanol production of ~13 mol L$^{-1}$ h$^{-1}$ at room temperature, with cogeneration of potassium formate and electricity. Godoi [21] explained that the effect was due to the affinity of methane for PdO and copper oxides activating water for the generation of reactive oxygenated species, such as the ●OH radical.

The association of Pd with ATO favors the formation of reactive oxygen species, according to Sun and coauthors [20], due to the synergistic effect of the adsorption of hydrogen from water at Pd sites. With the ease of tin and antimony oxides in breaking the H-OH bond, ATO also has good electrical conduction and corrosion resistance [21]. These characteristics make $SnO_2.Sb_2O_5$ is a great option for catalyst support for this reaction, Furthermore, it is added that carbon, usually used as catalyst support in this type of reactor, suffers from wear due to the same species generated by the activation of water, thus having its useful life shortened [22,23].

In this context, the application of the ATO becomes even more advantageous due to its synergic effect on the catalytic activity. In this work, the conversion of methane to methanol on metallic compound of the Pd and Cu catalysts in different proportions (Pd:Cu) supported on ATO was studied in a fuel cell polymer electrolyte reactor. The methanol formed was quantified by the HPLC (high-pressure liquid chromatograph) technique, and consequently, the reaction rate can be determined to portray the most active catalyst for converting methane to methanol in the fuel cell reactor.

## 2. Results

X-ray diffraction patterns of palladium and copper-based materials supported on ATO are presented in Figure 1. In this figure, it is possible to clearly observe the peaks in 2θ at ~26°, 33°, 38°, 51°, 54°, 62° and 64°, which corresponds to the support of the electrocatalysts $Sn_2O.Sb_2O_5$ [24]. The Scherer equation used to calculate the average crystallite size and these results indicated that larger crystals have narrower and more intense peaks, exactly the behavior observed for the crystallographic faces of the ATO that is predominant in all diffractograms.

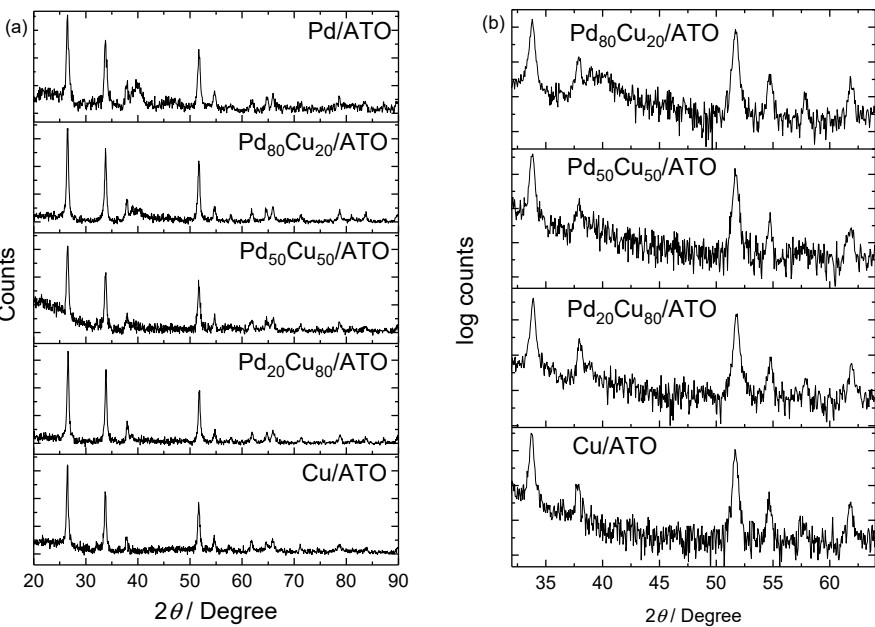

**Figure 1.** (**a**) X ray diffractogram pattern of the Pd and Cu catalysts supported on ATO; (**b**) catalysts doped with Cu X ray diffractogram pattern with the intensities normalized by the logarithmic function.

It is possible to observe well-widened Pd-related peaks at 2 θ ∼ 40°, 46°, 67°, and 82° associated with the crystalline planes (111), (200), (220), (311), and (222), according to (JCPDS# 89-4897). It is noted that these peaks practically disappear in the $Pd_{20}Cu_{80}$/ATO composition. However, the observation of copper-related peaks cannot be done due to the low crystallinity of the oxides of this metal. To observe the peaks related to copper, in Figure 1b, the logarithms of the intensities of the same diffractograms are presented, where it is possible to observe the peaks at 2 θ ∼ 36°, 38°, and 53°, referring to CuO (JCPDS# 45-0937).

The morphology of the nanoparticles was observed by transmission electrons microscopy, and Figure 2 portrays the micrographs and their respective particle size histograms. The average particle size measured was 9.81, 9.32, 9.27, 8.36, and 10.25 nm, respectively, for Pd/ATO, $Pd_{80}Cu_{20}$/ATO $Pd_{50}Cu_{50}$/ATO, $Pd_{20}Cu_{80}$/ATO, and Cu/ATO, which are higher than those reported for PdCu materials obtained by the same method supported on carbon [25], indicating that the support may have favored agglomeration and, in turn, the particle size.

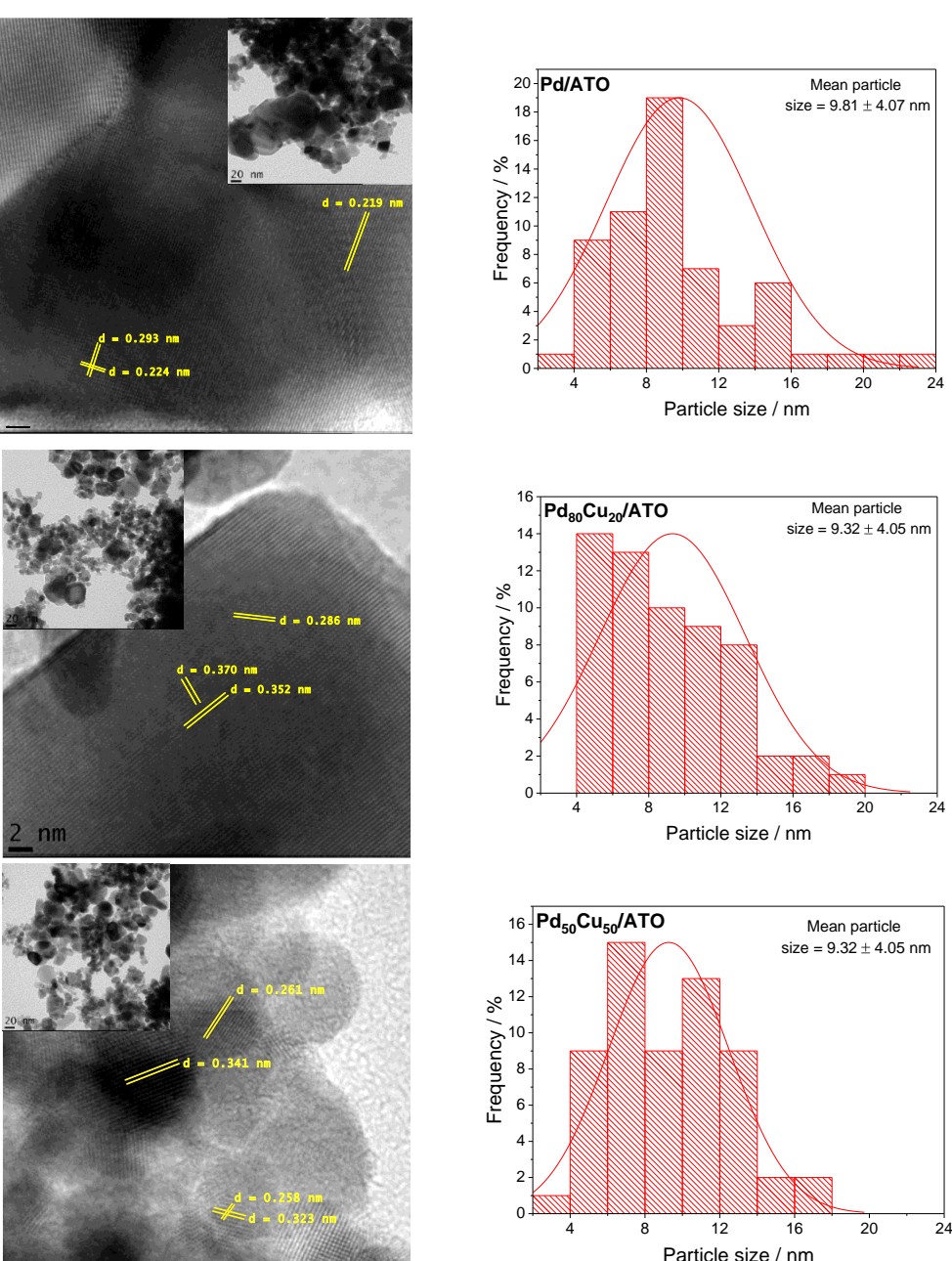

**Figure 2.** *Cont.*

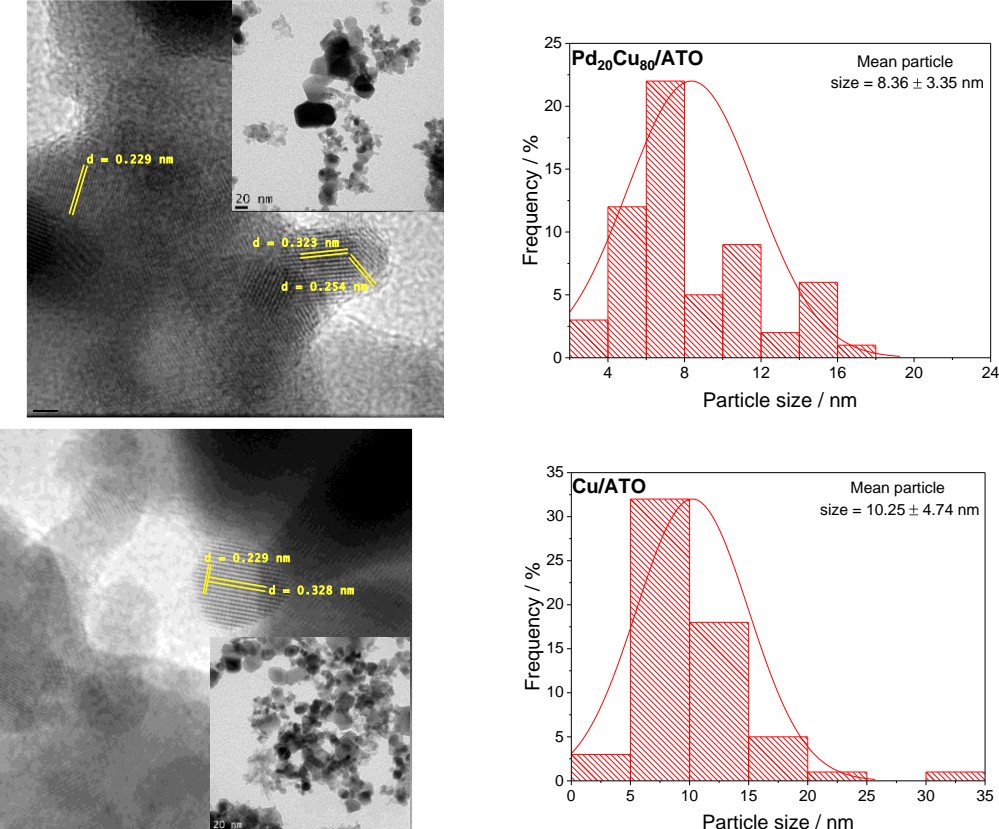

**Figure 2.** TEM micrograph and histograms of the particle size distribution to Pd and Cu catalysts supported on ATO.

It is observable that the catalytic nanoparticles are agglomerated in some regions of the oxide-based support, and this type of morphology was also observed for other nanoparticles supported on metallic oxides, such as ATO and $TiO_2$ [21,26,27]. Additionally, the nanoparticles composed of Pd and Cu had a high degree of crystallinity. It is possible to observe planes with distances of 0.438, 0.312–0.340, and 0.220–0.250 nm, characteristic of nanoparticles of metallic alloys supported on ATO, similar to that reported by Qu et al. [21].

Figure 3a portrays the cyclic voltammograms of the Pd:Cu catalysts, and it is possible to see the characteristic adsorption peaks of hydrogen desorption on palladium in the region from −0.85 to −0.6 V for Pd/ATO [25]; this process loses much definition with the addition of copper in the material, likely through a synergistic process of the oxides present that inhibit the presence of hydrogen on the palladium surface or covered the palladium nanoparticles [21]. Also, the addition of copper in the catalyst composition caused anodic peaks at −0.5 and −0.2 V, linked to the oxidation process of $Cu_2O/Cu(OH)_2$ [28,29] and $Cu/Cu_2O$ [28], respectively.

To obtain more details of the processes that occur in the interaction of the catalyst with water in 1.0 mol $L^{-1}$ NaOH aqueous solution, an important step for the partial oxidation of methane, in situ Raman-assisted electrochemical measurements, were performed as portrayed in Figure 3b–f. Where it is possible to observe the bands at 794, 974, and 1166 $cm^{-1}$ corresponding to ν(C–S), νs(C–O–C), and ν ($CF_2$) from Nafion [30], the bands at 489 related to the $E_g$ of $SnO_2$ and the convolved band with centers at 785 and ~831 $cm^{-1}$, related to the ATO [31,32]; all these bands present a behavior constantly.

The increase in the band centered at 639 $cm^{-1}$ corresponds to the bending of the PdO-H bond [33], indicating the oxidation of Pd as a function of potential, in the Pd/ATO catalyst. However, in the other materials this band was much less intense, corroborating what was observed in cyclic voltammetry, where the Pd profile is suppressed, likely due to the noble metal being covered by the less noble metal oxides.

In materials containing copper it was possible to observe a band at ~343 cm$^{-1}$ corresponding to CuO [34], without the appearance of bands related to Cu$_2$O, which is in agreement with the phase observed in the XRD, Figure 1b. This band becomes more evident in materials with higher copper content. The increase in the copper content in the catalyst also shifts the potential for the CuO band to provide more negative potentials.

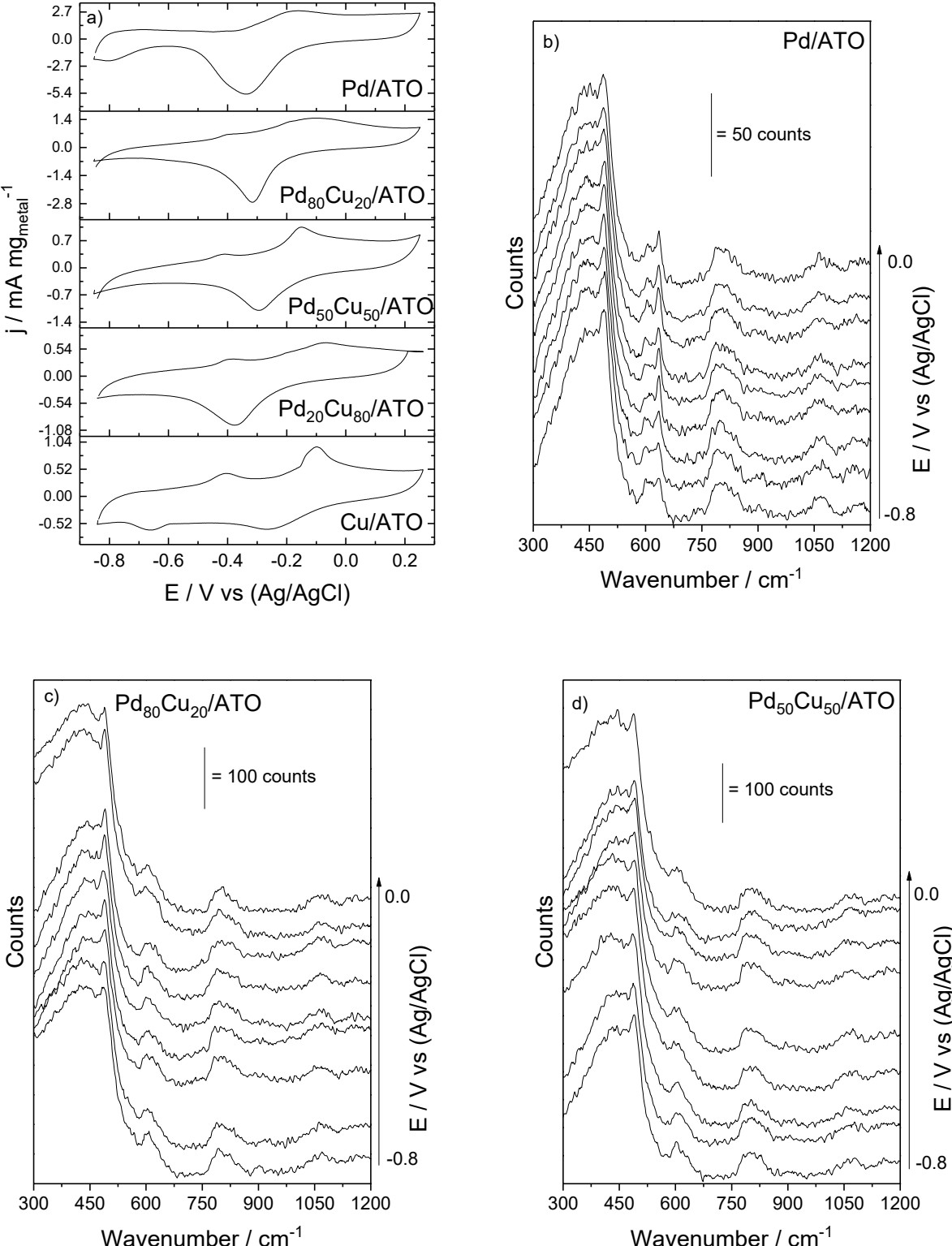

**Figure 3.** *Cont.*

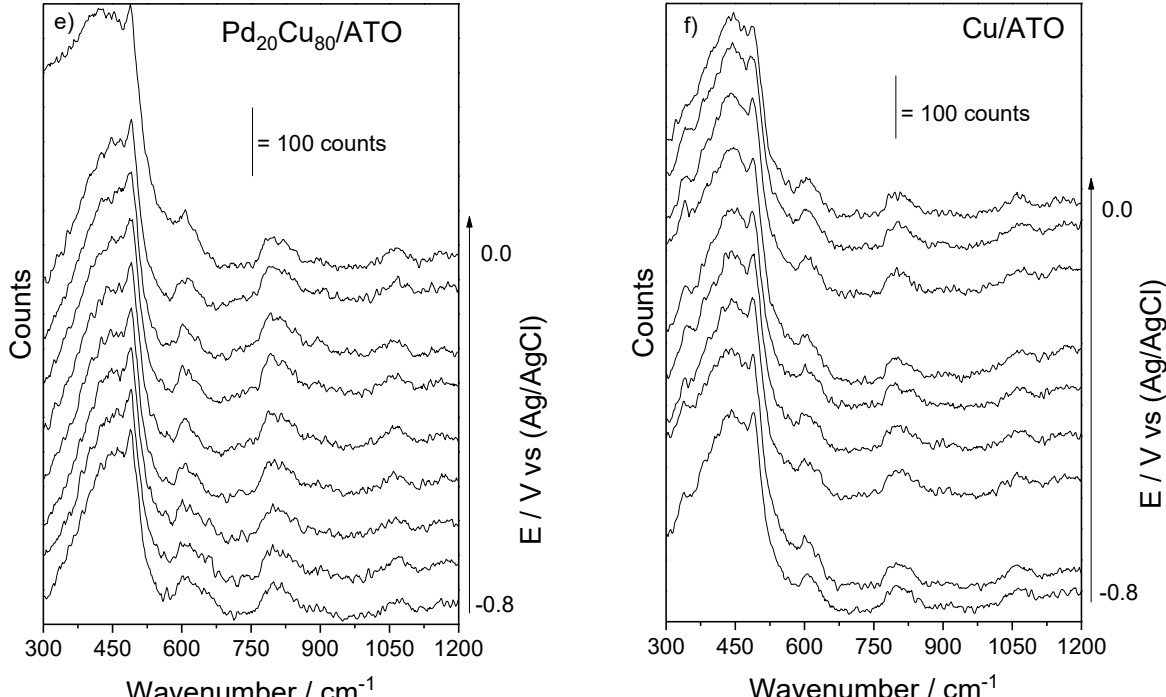

**Figure 3.** (**a**) Cyclic voltammetry curves of the PdCu electrodes in different proportions (scan rate v = 10 mV s$^{-1}$) in 1 mol L$^{-1}$ NaOH aqueous solution; (**b**–**f**) in-situ Raman spectra collected in same conditions collected at 100 mV.

The methane to methanol conversion reaction was performed in a polymeric electrolyte reactor—type fuel cell with co-generation of energy (Figure 4), and it was observed that the all materials present an open circuit value (OCV) very close to 0.45 V, indicating that ATO support demonstrates a predominant influence in this aspect. The OCV measured was about 50 mV higher than reported for similar reactors in the literature [35].

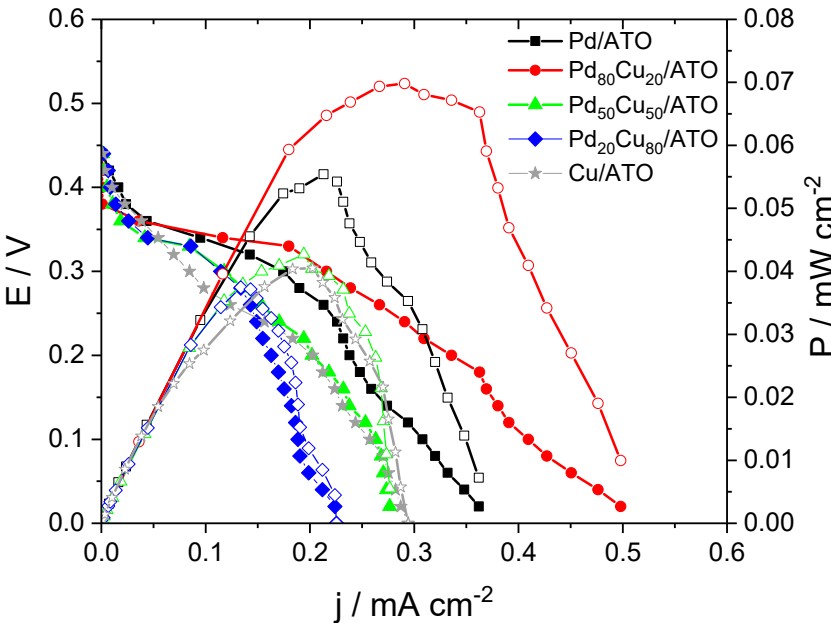

**Figure 4.** The polarization curves and the power density of a PdCu/ATO catalysts anode (5 mg cm$^{-2}$ catalysts loading) and Pt/C Basf or the cathode in all experiments (1.0 mg Pt cm$^{-2}$ catalyst loading with 20 wt% Pt loading on carbon), Nafion 117 membrane KOH treatment; NaOH 1.0 mol L$^{-1}$ + CH$_4$ 50 mL min$^{-1}$, and O$_2$ flux of 200 mL min$^{-1}$.

For $Pd_{80}Cu_{20}$/ATO (0.07 mW cm$^{-2}$) the maximum power density is about 26 % higher than Pd/ATO (0.05 mW cm$^{-2}$), the second material more active. For $Pd_{50}Cu_{50}$/ATO, $Pd_{20}Cu_{80}$/ATO, and Cu/ATO, the maximum power density obtained was, respectively 0.04, 0.038, and 0.04 mW cm$^{-2}$. Aliquots were collected from the reactor effluent every 100 mV for 5 min. The products were identified using infrared spectroscopy (Figure 5), where it was possible to observe bands referring to methanol at 1075 and 1032 cm$^{-1}$ [36,37], and they were visible when using the Pd/ATO, $Pd_{80}Cu_{20}$/ATO, and Cu/ATO materials, and for Cu/ATO it has a decreasing profile along with the potential.

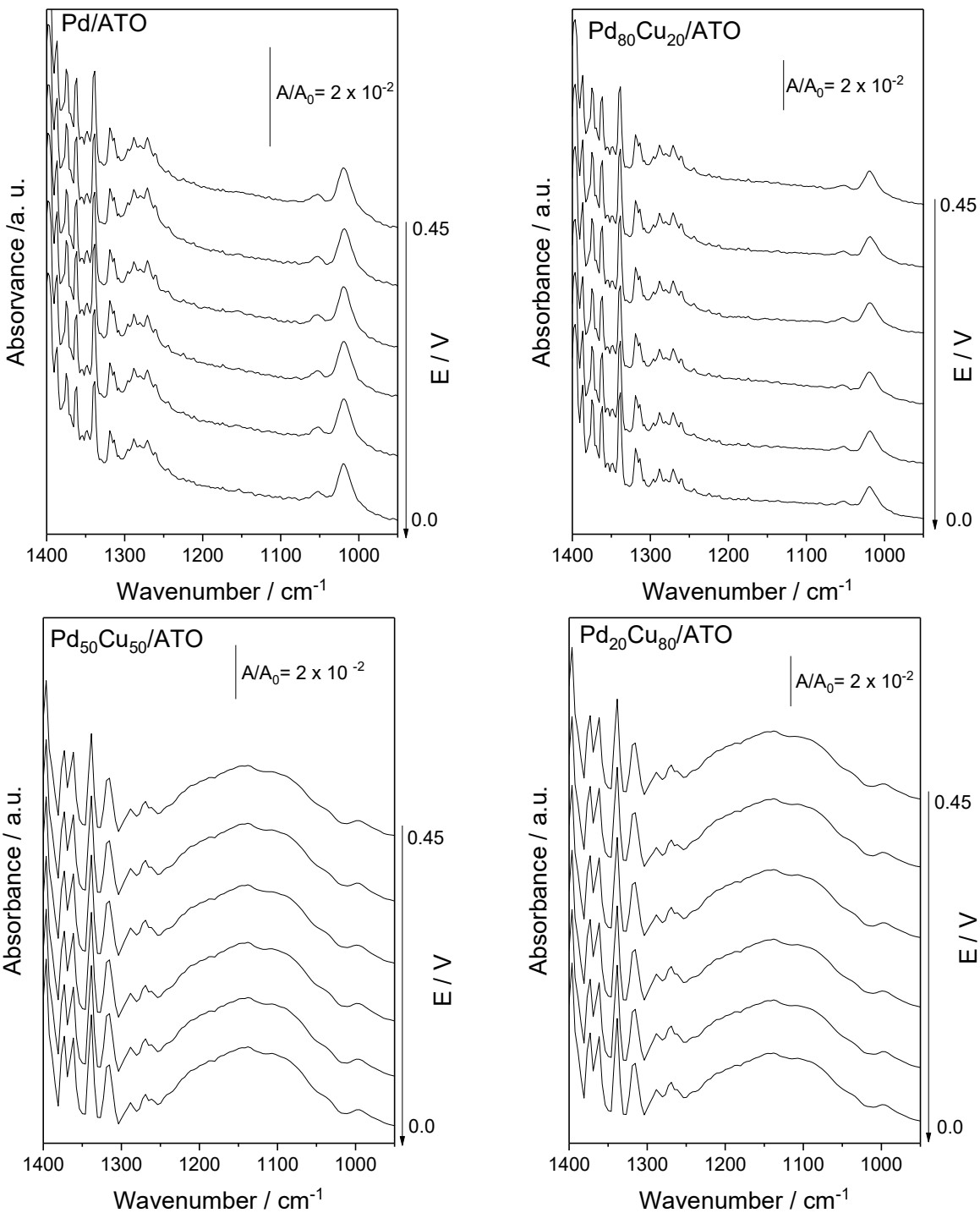

**Figure 5.** *Cont.*

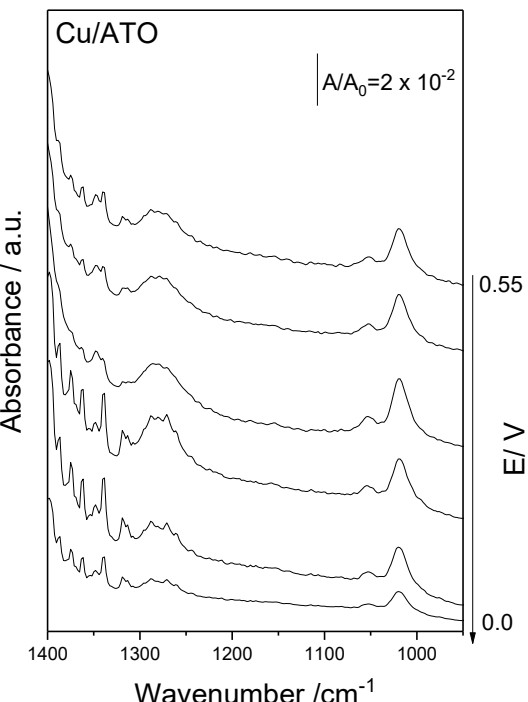

**Figure 5.** FTIR spectra of the aliquots were collected from the reactor effluent every 100 mV for 5 min.

For the materials $Pd_{50}Cu_{50}$/ATO and $Pd_{20}Cu_{80}$/ATO, it was possible to see the band referring to carbonate ions at 1375 cm$^{-1}$ [38,39], indicating that the association of Pd with high amounts of copper can promote more oxidized species. This feature was also marked by the wide band centered at 1140 cm$^{-1}$, indicating large proportions of sodium formate, confirmed by the narrow band at 1345 cm$^{-1}$ [40] which was also well distinguishable in other materials.

The methanol produced was quantified by liquid chromatography and reported as a rate reaction according to Equation (5) and the result is provided in Figure 6, where it is observable that the $Pd_{80}Cu_{20}$/ATO composition is the most active for the conversion of methane into methanol at all potentials.

$$r = \frac{Methanol_{amount}}{Volume \times Time} \tag{5}$$

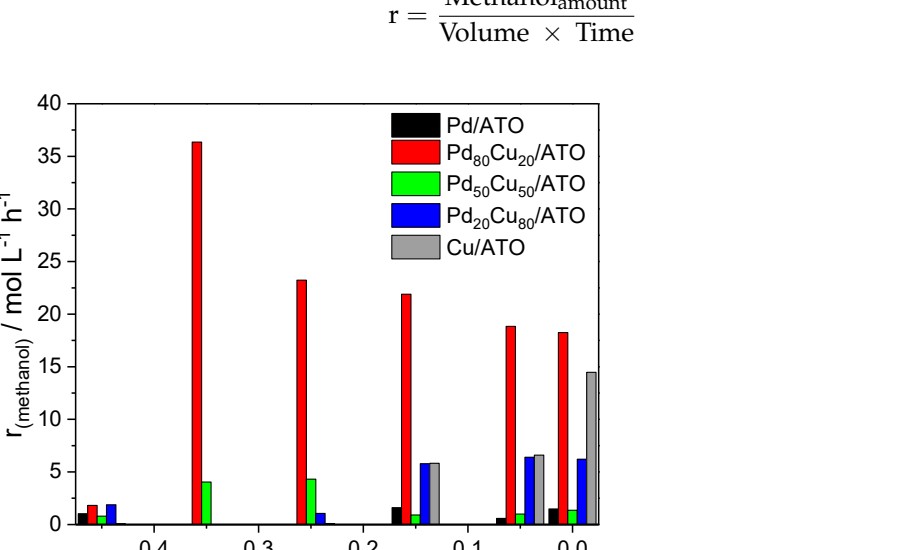

**Figure 6.** Rate reaction of all PdCu/ATO catalysts to methanol production in the liquid phase.

The use of ATO as a support favored the production of methanol from methane in Cu/ATO, which is practically not observed for Cu/C by Godoi et al. [25]. However, mixtures of Pd and Cu with high copper content seem to direct the production of more oxidized compounds, such as carbonate and formate.

## 3. Materials and Methods

The ATO-supported Pd, Cu-based materials were prepared by the $NaBH_4$ reduction method. This methodology consists of a mixture of ultrapure water with isopropanol 50/50 (*v/v*) with appropriate amount of ATO ($Sb_2O_5.SnO_2$—Aldrich St. Louis, MI, USA) and metallic precursors of $Pd(NO_3)_2.2H_2O$ (Aldrich) and $CuCl_2.2H_2O$ (Aldrich) to obtain a load of 20% by mass of these metals in relation to the ATO support. In this medium, $NaBH_4$ (Aldrich) was added in an aqueous solution with an excess of 5:1 in relation to the metals, and stirring was maintained for 30 min, after which time the material obtained was washed with ultrapure water and filtered [41].

The materials were characterized by X-ray diffraction, using a Rigaku X-ray diffractometer—Miniflex II, with a Cu$k\alpha$ radiation source of 0.15406Å. The diffractograms were obtained from 20° to 90°, with a scan speed of 2° $min^{-1}$. The morphology was observed by transmission electron microscopy performed by a transmission electron microscope JEOL JEM-2100, operated at 200 KeV. For the construction of the histogram and the calculation of the average size, 300 nanoparticles of each catalyst were digitally mediated from microscopy.

The catalysts were characterized by cyclic voltammetry performed in a three-electrode cell in a PGStat 302N Autolab potentiostat. The working electrode built on a glassy carbon support covered with an ultra-thin porous layer, produced from a paint made with 8 mg of catalyst, 600 μL of ultrapure water, 400 μL of isopropanol, and 15 μL of Nafion® (D-520) mixed in ultrasound. The Ag/AgCl 3.0 mol $L^{-1}$ electrode was used as a reference electrode and a 2 $cm^2$ Pt electrode was used as a counter-electrode. This same potentiostat and electrodes were used in conjunction with a Raman Macroram spectrometer—Horiba, with a 785 nm laser and an electrochemical cell suitable for performing the electrochemical assays assisted by in situ Raman spectroscopy.

The conversion of methane to methanol was performed in a polymeric electrolyte reactor—fuel cell type, with electrodes constructed with 1.0 mg of Pd + Cu per $cm^2$ at the anode, a membrane of Nafion 117 treated with KOH as electrolyte and 1.0 mg of Pt/C—Basf (20% *w/w*) as cathode. All electrodes were prepared by depositing the ink containing the catalyst with 30% by mass of a solution of Nafion D-520 (Aldrich) and isopropanol applied by brushing on a carbon cloth treated with PTFE. The reactor is based on a typical fuel cell coil plate design made in 316 L steel, fed with a mixture of methane 100 mL $min^{-1}$ and NaOH 1 mL $min^{-1}$, the ambient temperature at the anode, while the cathode is supplied with humidified $O_2$ in a bottle at a temperature of 85 °C with a flow rate of 400 mL $min^{-1}$ at the cathode.

Aliquots of the reactor effluent are collected every 100 mV for 300 s from the open circuit potential to 0V and analyzed by infrared spectroscopy performed on a Nicolett® 6700 with ATR Miracle (Pike) accessory and diamond/ZnSe crystal and a detector of MCT, and high-performance liquid chromatography (YL9100) with UV/Vis detector with detection made at 205 nm, with flow of 0.8 mL $min^{-1}$ of 50% water and 50% acetonitrile in an isocratic run on a C18 column (Phenomenex Luna 5 μm, 250 × 4.6 mm). The calibration curve presents the following linear equation: peaks area = 59.916 + 238.59 [*methanol*], and presenting $r^2$ = 0.9981.

## 4. Conclusions

The use of ATO as a catalyst support demonstrated promise in the partial oxidation reaction of methane to methanol, despite favoring the formation of nanoparticles larger than those supported on carbon. The TEM images portray the well-defined planes corresponded to the metal nanoparticles supported on the ATO surface. It is possible that the use of $SnO_2.Sb_2O_5$ as a support also inhibits the formation of Cu2O in the catalyst, a very common

phase, and this mixture of oxides alters the properties of Pd in hydrogen adsorption. However, this fact does not decrease the activity to produce methanol from methane, as observed for Pd$_{80}$Cu$_{20}$/ATO.

**Author Contributions:** C.M.G., planning and execution of research; I.M.G., execution of research; P.V.R.G., execution of measurements; J.F.C., planning and execution of HPLC measurements; P.J.Z., planning, discussing, and writing the draft; L.O., execution of TEM experiments; A.O.N., oversight and leadership responsibility for the research activity planning and execution; R.F.B.d.S., management and coordination for the research activity planning and writing of the final draft. All authors have read and agreed to the published version of the manuscript.

**Funding:** This research was funded by CAPES, CNPq (302709/2020-7), FAPESP (2017/11937-4) and CINE-SHELL (ANP).

**Institutional Review Board Statement:** Not applicable.

**Informed Consent Statement:** Not applicable.

**Conflicts of Interest:** The authors declare no conflict of interest.

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
