# Peer review of "Production of Methanol on PdCu/ATO in a Polymeric Electrolyte Reactor of the Fuel Cell Type from Methane"

_methane, doi:10.3390/methane1030018_

Round 1
Reviewer 1 Report
After I read this manuscript carefully, the author tried to develop Pd/Cu@ATO as the electrocatalysts for the conversion of methane into methanol. It is very important for further applications. However, there are a lot of questions that are not clear. Here are my major concerns:
1. The writing of this manuscript has to be improved. The authors have to correct the spelling and grammar errors carefully and the manuscript has also checked by a native English speaker.
2. In the introduction, the authors only reported someone prepared the catalysts and obtained the high conversion efficiency of methane to methanol. However, it is no sense for a general readers. It is necessary to show the detail results such as conversion efficiency, selectivity or any others. Please add them in the manuscript.
3. The peaks in the Figure 2 are very difficult for me to understand. It is necessary to show the crystal plane and the phase for each peak. Also, the figure 1 is missing or the typing error for figure caption.
4. The quality of some TEM images is low. For example, the d spaces of Pd/ATO, Pd80Cu20/ATO are hard to observed. Please show the TEM images with high resolution.
5.For the CV curve, the authors just reported the hydrogen absorption peak at the applied voltage of -0.6 - -0.85 V vs. Ag/AgCl using PdCu catalysts. For the Pd/ATO sample, what is the reaction at the applied voltage of -0.6 - -0.2 V using Pd/ATO, I think it is different for the Cu/ATO, or Pd-Cu/ATO, the authors have discussed them carefully.
6.I suggest the author also check the composition of gas phase in order to examine the selectivity of methane reaction and also the conversion of methane. The methane may change into other products rather than methanol.
7.How about the increase of decrease of methane in the reaction using their reaction. Did the conversion efficiency of selectivity of reaction change with the change of methane flow rate.
Author Response
- The writing of this manuscript has to be improved. The authors have to correct the spelling and grammar errors carefully and the manuscript has also checked by a native English speaker.
A: Was done
- In the introduction, the authors only reported someone prepared the catalysts and obtained the high conversion efficiency of methane to methanol. However, it is no sense for a general readers. It is necessary to show the detail results such as conversion efficiency, selectivity or any others. Please add them in the manuscript.
A: Was improve the introduction
- The peaks in the Figure 2 are very difficult for me to understand. It is necessary to show the crystal plane and the phase for each peak. Also, the figure 1 is missing or the typing error for figure caption.
A: Unfortunately, we made a mistake in the captions of figure 1 referring to XRD, already corrected. However, as for indicating faces and phases in the figures, it was the authors' choice not to do so, because the indications can interfere with the visualization of the measured data. As described in the text we have at least 4 phases present in the diffractograms and each one with several faces, and these faces of different phases sometimes convolute creating a figure that is difficult to identify the data, for this reason the option to describe the faces and phases in the text.
- The quality of some TEM images is low. For example, the d spaces of Pd/ATO, Pd80Cu20/ATO are hard to observed. Please show the TEM images with high resolution.
A: The images have been replaced by high-resolution images with the planes of the Pd metals and the PdCu material highlighted for clarity.
- For the CV curve, the authors just reported the hydrogen absorption peak at the applied voltage of -0.6 - -0.85 V vs. Ag/AgCl using PdCu catalysts. For the Pd/ATO sample, what is the reaction at the applied voltage of -0.6 - -0.2 V using Pd/ATO, I think it is different for the Cu/ATO, or Pd-Cu/ATO, the authors have discussed them carefully.
A: in region of -0.6 and -0.2 for Pd/ATO is reported as a region of influence of electric doble layer until the onset potential for Pd and ATO ~ -0.3V, being this fact not configuring a fingerprint of the material, so it was not commented on in the text.
- I suggest the author also check the composition of gas phase in order to examine the selectivity of methane reaction and also the conversion of methane. The methane may change into other products rather than methanol.
A: this suggestion can be adopted in future works, however, in other works by the group, hydrogen, ethane and propane have already been found in the gas phase, however in very small amounts
- How about the increase of decrease of methane in the reaction using their reaction. Did the conversion efficiency of selectivity of reaction change with the change of methane flow rate.
A: The methane flux interferes with selectivity and activity, larger flows decrease activity
however, smaller methane fluxes favor the formation of more oxidized products such as carbonates and formate. However, this operational parameter is not part of the scope of this work.
Reviewer 2 Report
The abstract should contain some important outcomes including some numerical values of the current research.
Try to avoid mentioning the methods of characterization followed. It will be better to visualize the observed results from such characterizations.
Results are given after introduction. Plz, check the format of the journal.
The sub and super script should be well defined in the text.
Many authors are there in the submission, so mention the individual contributions in the declaration section.
Recently published papers should be cited in the manuscript, https://doi.org/10.1016/j.chemosphere.2022.134299, https://doi.org/10.1007/s13399-021-01872-5.
Cite some papers which are published in this journal.
How the present study is different as reported earlier is missing in the introduction section.
Apart from this the manuscript is explained very well.
Author Response
The abstract should contain some important outcomes including some numerical values of the current research. Try to avoid mentioning the methods of characterization followed. It will be better to visualize the observed results from such characterizations. Results are given after introduction. Plz, check the format of the journal. The sub and super script should be well defined in the text. Many authors are there in the submission, so mention the individual contributions in the declaration section.
Recently published papers should be cited in the manuscript, https://doi.org/10.1016/j.chemosphere.2022.134299, https://doi.org/10.1007/s13399-021-01872-5. Cite some papers which are published in this journal. How the present study is different as reported earlier is missing in the introduction section. Apart from this the manuscript is explained very well.
A: the article has been rewritten in some parts to accommodate the suggestions. The references suggested we cited the Conversion of methane to methanol: technologies and future challenges, this work have informations that contributed with the present manuscript.
Round 2
Reviewer 1 Report
In this revised manuscript, the authors have tried to answer the reviewer's question. However, I still have one suggestion. Because the authors only detected the selectivity and reaction rate in the liquid phase, the title and figure caption of figure 6 have to been modified for the reaction rate in the liquid phase.
Author Response
Q: In this revised manuscript, the authors have tried to answer the reviewer's question. However, I still have one suggestion. Because the authors only detected the selectivity and reaction rate in the liquid phase, the title and figure caption of figure 6 have to been modified for the reaction rate in the liquid phase.
A: was done